# Adaptive Smoothed Online Multi-Task Learning

**Keerthiram Murugesan**[*]
Carnegie Mellon University
kmuruges@cs.cmu.edu

**Hanxiao Liu**[*]
Carnegie Mellon University
hanxiaol@cs.cmu.edu

**Jaime Carbonell**
Carnegie Mellon University
jgc@cs.cmu.edu

**Yiming Yang**
Carnegie Mellon University
yiming@cs.cmu.edu

## Abstract

This paper addresses the challenge of jointly learning both the per-task model parameters and the inter-task relationships in a multi-task online learning setting. The proposed algorithm features probabilistic interpretation, efficient updating rules and flexible modulation on whether learners focus on their specific task or on jointly address all tasks. The paper also proves a sub-linear regret bound as compared to the best linear predictor in hindsight. Experiments over three multi-task learning benchmark datasets show advantageous performance of the proposed approach over several state-of-the-art online multi-task learning baselines.

## 1 Introduction

The power of joint learning in multiple tasks arises from the transfer of relevant knowledge across said tasks, especially from information-rich tasks to information-poor ones. Instead of learning individual models, multi-task methods leverage the relationships between tasks to jointly build a better model for each task. Most existing work in multi-task learning focuses on how to take advantage of these task relationships, either to share data directly [1] or to learn model parameters via cross-task regularization techniques [2, 3, 4]. In a broad sense, there are two settings to learn these task relationships 1) batch learning, in which an entire training set is available to the learner 2) online learning, in which the learner sees the data in a sequential fashion. In recent years, online multi-task learning has attracted extensive research attention [5, 6, 7, 8, 9].

Following the online setting, particularly from [6, 7], at each round $t$, the learner *receives* a set of $K$ observations from $K$ tasks and *predicts* the output label for each of these observations. Subsequently, the learner receives the true labels and *updates* the model(s) as necessary. This sequence is repeated over the entire data, simulating a data stream. Our approach follows an error-driven update rule in which the model for a given task is updated only when the prediction for that task is in error. The goal of an online learner is to minimize errors compared to the full hindsight learner. The key challenge in online learning with large number of tasks is to adaptively learn the model parameters and the task relationships, which potentially change over time. Without manageable efficient updates at each round, learning the task relationship matrix automatically may impose a severe computational burden. In other words, we need to make predictions and update the models in an efficient real time manner.

We propose an online learning framework that *efficiently* learns multiple related tasks by estimating the task relationship matrix from the data, along with the model parameters for each task. We learn the model for each task by sharing data from related task directly. Our model provides a natural way to specify the trade-off between learning the hypothesis from each task's own (possibly quite

---

[*]Both student authors contributed equally.

limited) data and data from multiple related tasks. We propose an iterative algorithm to learn the task parameters and the task-relationship matrix alternatively. We first describe our proposed approach under a batch setting and then extend it to the online learning paradigm. In addition, we provide a theoretical analysis for our online algorithm and show that it can achieve a sub-linear regret compared to the best linear predictor in hindsight. We evaluate our model with several state-of-the-art online learning algorithms for multiple tasks.

There are many useful application areas for online multi-task learning, including optimizing financial trading, email prioritization, personalized news, and spam filtering. Consider the latter, where some spam is universal to all users (e.g. financial scams), some messages might be useful to certain affinity groups, but spam to most others (e.g. announcements of meditation classes or other special interest activities), and some may depend on evolving user interests. In spam filtering each user is a task, and shared interests and dis-interests formulate the inter-task relationship matrix. If we can learn the matrix as well as improving models from specific spam/not-spam decisions, we can perform mass customization of spam filtering, borrowing from spam/not-spam feedback from users with similar preferences. The primary contribution of this paper is precisely the joint learning of inter-task relationships and its use in estimating per-task model parameters in an online setting.

## 1.1 Related Work

While there is considerable literature in online multi-task learning, many crucial aspects remain largely unexplored. Most existing work in online multi-task learning focuses on how to take advantage of task relationships. To achieve this, Lugosi et. al [7] imposed a hard constraint on the $K$ simultaneous actions taken by the learner in the expert setting, Agarwal et. al [10] used matrix regularization, and Dekel et. al [6] proposed a global loss function, as an absolute norm, to tie together the loss values of the individual tasks. Different from existing online multi-task learning models, our paper proposes an intuitive and efficient way to learn the task relationship matrix automatically from the data, and to explicitly take into account the learned relationships during model updates.

Cavallanti et. al [8] assumes that task relationships are available *a priori*. Kshirsagar et. al [11] does the same but in a more adaptive manner. However such task-relation prior knowledge is either unavailable or infeasible to obtain for many applications especially when the number of tasks $K$ is large [12] and/or when the manual annotation of task relationships is expensive [13]. Saha et. al [9] formulated the learning of task relationship matrix as a Bregman-divergence minimization problem w.r.t. positive definite matrices. The model suffers from high computational complexity as semi-definite programming is required when updating the task relationship matrix at each online round. We show that with a different formulation, we can obtain a similar but much cheaper updating rule for learning the inter-task weights.

The most related work to ours is *Shared Hypothesis* model (*SHAMO*) from Crammer and Mansour [1], where the key idea is to use a K-means-like procedure that simultaneously clusters different tasks and learns a small pool of $m \ll K$ shared hypotheses. Specifically, each task is free to choose a hypothesis from the pool that better classifies its own data, and each hypothesis is learned from pooling together all the training data that belongs to the same cluster. A similar idea was explored by Abernathy et. al [5] under expert settings.

## 2 Smoothed Multi-Task Learning

### 2.1 Setup

Suppose we are given $K$ tasks where the $j^{th}$ task is associated with $N_j$ training examples. For brevity we consider a binary classification problem for each task, but the methods generalize to multi-class and are also applicable to regression tasks. We denote by $[N]$ the consecutive integers ranging from 1 to $N$. Let $\left\{(x_j^{(i)}, y_j^{(i)})\right\}_{i=1}^{N_j}$ and $L_j(w) = \frac{1}{N_j} \sum_{i \in [N_j]} \left(1 - y_j^{(i)} \langle x_j^{(i)}, w \rangle\right)_+$ be the training set and batch empirical loss for task $j$, respectively, where $(z)_+ = \max(0, z)$, $x_j^{(i)} \in \mathbb{R}^d$ is the $i^{th}$ instance from the $j^{th}$ task and $y_j^{(i)}$ is its corresponding true label.

We start from the motivation of our formulation in Section 2.2, based on which we first propose a batch formulation in Section 2.3. Then, we extend the method to the online setting in Section 2.4.

## 2.2 Motivation

Learning tasks may be addressed independently via $w_k^* = \mathrm{argmin}_{w_k} L_k(w_k), \forall k \in [K]$. However, when each task has limited training data, it is often beneficial to allow information sharing among the tasks, which can be achieved via the following optimization:

$$w_k^* = \mathrm{argmin}_{w_k} \sum_{j \in [K]} \eta_{kj} L_j(w_k) \quad \forall k \in [K] \tag{1}$$

Beyond each task $k$, optimization (1) encourages hypothesis $w_k^*$ to do well on the remaining $K - 1$ tasks thus allowing tasks to borrow information from each other. In the extreme case where the $K$ tasks have an identical data distribution, optimization (1) amounts to using $\sum_{j \in [K]} N_j$ examples for training as compared to $N_k$ in independent learning.

The weight matrix $\boldsymbol{\eta}$ is in essence a task relationship matrix, and a prior may be manually specified according to domain knowledge about the tasks. For instance, $\eta_{kj}$ would typically be set to a large value if tasks $k$ and $j$ share similar nature. If $\boldsymbol{\eta} = \boldsymbol{I}$, (1) reduces to learning tasks independently. It is clear that manual specification of $\boldsymbol{\eta}$ is feasible only when $K$ is small. Moreover, tasks may be statistically correlated even if a domain expert is unavailable to identify an explicit relation, or if the effort required is too great. Hence, it is often desirable to automatically estimate the optimal $\boldsymbol{\eta}$ adapted to the inter-task problem structure.

We propose to learn $\eta$ in a data-driven manner. For the $k^{th}$ task, we optimize

$$w_k^*, \eta_k^* = \mathrm{argmin}_{w_k, \eta_k \in \Theta} \sum_{j \in [K]} \eta_{kj} L_j(w_k) + \lambda r(\eta_k) \tag{2}$$

where $\Theta$ defines the feasible domain of $\eta_k$, and regularizer $r$ prevents degenerate cases, e.g., where $\eta_k$ becomes an all-zero vector. Optimization (2) shares the same underlying insight with Self-Paced Learning (SPL) [14, 15] where the algorithm automatically learns the weights over data points during training. However, the process and scope in the two methods differ fundamentally: SPL minimizes the weighted loss over *datapoints* within a single domain, while optimization (2) minimizes the weighted loss over multiple *tasks* across possibly heterogeneous domains.

A common choice of $\Theta$ and $r(\eta_k)$ in SPL is $\Theta = [0, 1]^K$ and $r(\eta_k) = -\|\eta_k\|_1$. There are several drawbacks of naively applying this type of settings to the multitask scenario: (i) *Lack of focus*: there is no guarantee that the $k^{th}$ learner will put more focus on the $k^{th}$ task itself. When task $k$ is intrinsically difficult, $\eta_{kk}^*$ could simply be set near zero and $w_k^*$ becomes almost independent of the $k^{th}$ task. (ii) *Weak interpretability*, the learned $\eta_k^*$ may not be interpretable as it is not directly tied to any physical meanings (iii) *Lack of worst-case guarantee* in the online setting. All those issues will be addressed by our proposed model in the following.

## 2.3 Batch Formulation

We parametrize the aforementioned task relationship matrix $\boldsymbol{\eta} \in \mathbb{R}^{K \times K}$ as follows:

$$\boldsymbol{\eta} = \alpha \boldsymbol{I}_K + (1 - \alpha) \boldsymbol{P} \tag{3}$$

where $\boldsymbol{I}_K \in \mathbb{R}^{K \times K}$ is an identity matrix, $\boldsymbol{P} \in \mathbb{R}^{K \times K}$ is a *row-stochastic matrix* and $\alpha$ is a scalar in $[0, 1]$. Task relationship matrix $\boldsymbol{\eta}$ defined as above has the following interpretations:

1. *Concentration Factor* $\alpha$ quantifies the learners' "concentration" on their own tasks. Setting $\alpha = 1$ amounts to independent learning. We will see from the forthcoming Theorem 1 how to specify $\alpha$ to ensure the optimality of the online regret bound.

2. *Smoothed Attention Matrix* $\boldsymbol{P}$ quantifies to which degree the learners are attentive to all tasks. Specifically, define the $k^{th}$ row of $\boldsymbol{P}$, namely $p_k \in \Delta^{K-1}$, as a probability distribution over all tasks where $\Delta^{K-1}$ denotes the probability simplex. Our goal of learning a data-adaptive $\boldsymbol{\eta}$ now becomes learning a data-adaptive attention matrix $\boldsymbol{P}$.

Common choices about $\boldsymbol{\eta}$ in several existing algorithms are special cases of (3). For instance, domain adaptation assumes $\alpha = 0$ and a fixed row-stochastic matrix $\boldsymbol{P}$; in multi-task learning, we obtain the

effective heuristics of specifying $\boldsymbol{\eta}$ by Cavallanti et. al. [8] when $\alpha = \frac{1}{1+K}$ and $\boldsymbol{P} = \frac{1}{K}\boldsymbol{11}^\top$. When there are $m \ll K$ unique distributions $p_k$, then the problem reduces to *SHAMO* model [1].

Equation (3) implies the task relationship matrix $\boldsymbol{\eta}$ is also row-stochastic, where we always reserve probability $\alpha$ for the $k^{th}$ task itself as $\eta_{kk} \geq \alpha$. For each learner, the presence of $\alpha$ entails a trade off between learning from other tasks and concentrating on its own task. Note that we do not require $\boldsymbol{P}$ to be symmetric due to the asymmetric nature of information transferability—while classifiers trained on a resource-rich task can be well transferred to a resource-scarce task, the inverse is not usually true. Motivated by the above discussion, our batch formulation instantiates (2) as follows

$$w_k^*, p_k^* = \operatorname{argmin}_{w_k, p_k \in \Delta^{K-1}} \sum_{j \in [K]} \eta_{kj}(p_k) L_j(w_k) - \lambda \mathcal{H}(p_k) \tag{4}$$

$$= \operatorname{argmin}_{w_k, p_k \in \Delta^{K-1}} \mathbb{E}_{j \sim Multinomial(\eta_k(p_k))} L_j(w_k) - \lambda \mathcal{H}(p_k) \tag{5}$$

where $\mathcal{H}(p_k) = -\sum_{j \in [K]} p_{kj} \log p_{kj}$ denotes the entropy of distribution $p_k$. Optimization (4) can be viewed as to balance between minimizing the cross-task loss with mixture weights $\eta_k$ and maximizing the smoothness of cross-task attentions. The max-entropy regularization favours a uniform attention over all tasks and leads to analytical updating rules for $p_k$ (and $\eta_k$).

Optimization (4) is biconvex over $w_k$ and $p_k$. With $p_k^{(t)}$ fixed, solution for $w_k$ can be obtained using off-the-shelf solvers. With $w_k^{(t)}$ fixed, solution for $p_k$ is given in closed-form:

$$p_{kj}^{(t+1)} = \frac{e^{-\frac{1-\alpha}{\lambda} L_j(w_k^{(t)})}}{\sum_{j'=1}^{K} e^{-\frac{1-\alpha}{\lambda} L_{j'}(w_k^{(t)})}} \quad \forall j \in [K] \tag{6}$$

The exponential updating rule in (6) has an intuitive interpretation. That is, our algorithm attempts to use hypothesis $w_k^{(t)}$ obtained from the $k^{th}$ task to classify training examples in all other tasks. Task $j$ will be treated as related to task $k$ if its training examples can be well classified by $w_k$. The intuition is that two tasks are likely to relate to each other if they share similar decision boundaries, thus combining their associated data should yield to a stronger model, trained over larger data.

## 2.4 Online Formulation

In this section, we extend our batch formulation to the online setting. We assume that all tasks will be performed at each round, though the assumption can be relaxed with some added complexity to the method. At time $t$, the $k^{th}$ task receives a training instance $x_k^{(t)}$, makes a prediction $\langle x_k^{(t)}, w_k^{(t)} \rangle$ and suffers a loss after $y^{(t)}$ is revealed. Our algorithm follows a error-driven update rule in which the model is updated only when a task makes a mistake.

Let $\ell_{kj}^{(t)}(w) = 1 - y_j^{(t)} \langle x_j^{(t)}, w \rangle$ if $y_j^{(t)} \langle x_j^{(t)}, w_k^{(t)} \rangle < 1$ and $\ell_{kj}(w) = 0$ otherwise. For brevity, we introduce shorthands $\ell_{kj}^{(t)} = \ell_{kj}^{(t)}(w_k^{(t)})$ and $\eta_{kj}^{(t)} = \eta_{kj}(p_k^{(t)})$.

For the $k^{th}$ task we consider the following optimization problem at each time:

$$w_k^{(t+1)}, p_k^{(t+1)} = \operatorname*{argmin}_{w_k, p_k \in \Delta^{K-1}} C \sum_{j \in [K]} \eta_{kj}(p_k) \ell_{kj}^{(t)}(w_k) + \|w_k - w_k^{(t)}\|^2 + \lambda D_{\mathrm{KL}}(p_k \| p_k^{(t)}) \tag{7}$$

where $\sum_{j \in [K]} \eta_{kj}(p_k) \ell_{kj}^{(t)}(w_k) = \mathbb{E}_{j \sim Multi(\eta_k(p_k))} \ell_{kj}^{(t)}(w_k)$, and $D_{\mathrm{KL}}(p_k \| p_k^{(t)})$ denotes the Kullback–Leibler (KL) divergence between current and previous soft-attention distributions. The presence of last two terms in (7) allows the model parameters to evolve smoothly over time. Optimization (7) is naturally analogous to the batch optimization (4), where the batch loss $L_j(w_k)$ is replaced by its noisy version $\ell_{kj}^{(t)}(w_k)$ at time $t$, and negative entropy $-\mathcal{H}(p_k) = \sum_j p_{kj} \log p_{kj}$ is replaced by $D_{\mathrm{KL}}(p_k \| p_k^{(t)})$ also known as the relative entropy. We will show the above formulation leads to analytical updating rules for both $w_k$ and $p_k$, a desirable property particularly as an online algorithm.

Solution for $w_k^{(t+1)}$ conditioned on $p_k^{(t)}$ is given in closed-form by the proximal operator

$$w_k^{(t+1)} = \mathbf{prox}(w_k^{(t)}) = \text{argmin}_{w_k} \ C \sum_{j \in [K]} \eta_{kj}(p_k^{(t)}) \ell_{kj}^{(t)}(w_k) + \|w_k - w_k^{(t)}\|^2 \tag{8}$$

$$= w_k^{(t)} + C \sum_{j: y_j^{(t)} \langle x_j^{(t)}, w_k^{(t)} \rangle < 1} \eta_{kj}(p_k^{(t)}) y_j^{(t)} x_j^{(t)} \tag{9}$$

Solution for $p_k^{(t+1)}$ conditioned on $w_k^{(t)}$ is also given in closed-form, analogous to mirror descent [16]

$$p_k^{(t+1)} = \text{argmin}_{p_k \in \Delta^{K-1}} \ C(1-\alpha) \sum_{j \in [K]} p_{kj} \ell_{kj}^{(t)} + \lambda D_{\text{KL}}\left(p_k \| p_k^{(t)}\right) \tag{10}$$

$$\implies p_{kj}^{(t+1)} = \frac{p_{kj}^{(t)} e^{-\frac{C(1-\alpha)}{\lambda} \ell_{kj}^{(t)}}}{\sum_{j'} p_{kj'}^{(t)} e^{-\frac{C(1-\alpha)}{\lambda} \ell_{kj'}^{(t)}}} \quad j \in [K] \tag{11}$$

The pseudo-code is in Algorithm 2[2]. Our algorithm is "passive" in the sense that updates are carried out only when a classification error occurs, namely when $\hat{y}_k^{(t)} \neq y_k^{(t)}$. An alternative is to perform "aggressive" updates only when the active set $\{j : y_j^{(t)} \langle x_j^{(t)}, w_k^{(t)} \rangle < 1\}$ is non-empty.

---

**Algorithm 1:** Batch Algorithm (*SMTL*-e)

**while** *not converge* **do**
    **for** $k \in [K]$ **do**
        $w_k^{(t)} \leftarrow \text{argmin}_{w_k} \ \alpha L_k(w_k) + (1 - \alpha) \sum_{j \in [K]} p_{kj}^{(t)} L_j(w_k)$;
        **for** $j \in [K]$ **do**
            $p_{kj}^{(t+1)} \leftarrow \dfrac{e^{-\frac{1-\alpha}{\lambda} L_j(w_k^{(t)})}}{\sum_{j'=1}^{K} e^{-\frac{1-\alpha}{\lambda} L_{j'}(w_k^{(t)})}}$;
        **end**
    **end**
    $t \leftarrow t + 1$;
**end**

---

**Algorithm 2:** Online Algorithm (*OSMTL*-e)

**for** $t \in [T]$ **do**
    **for** $k \in [K]$ **do**
        **if** $y_k^{(t)} \langle x_k^{(t)}, w_k^{(t)} \rangle < 1$ **then**
            $w_k^{(t+1)} \leftarrow w_k^{(t)} + C\alpha \mathbf{1}_{\ell_{kk}^{(t)} > 0} y_k^{(t)} x_k^{(t)} + C(1-\alpha) \sum_{j: \ell_{kj}^{(t)} > 0} p_{kj}^{(t)} y_j^{(t)} x_j^{(t)}$;
            **for** $j \in [K]$ **do**
                $p_{kj}^{(t+1)} \leftarrow \dfrac{p_{kj}^{(t)} e^{-\frac{C(1-\alpha)}{\lambda} \ell_{kj}^{(t)}}}{\sum_{j'=1}^{K} p_{kj'}^{(t)} e^{-\frac{C(1-\alpha)}{\lambda} \ell_{kj'}^{(t)}}}$;
            **end**
        **else**
            $w_k^{(t+1)}, p_k^{(t+1)} \leftarrow w_k^{(t)}, p_k^{(t)}$;
        **end**
    **end**
**end**

---

## 2.5 Regret Bound

**Theorem 1.** $\forall k \in [K]$, let $S_k = \left\{\left(x_k^{(t)}, y_k^{(t)}\right)\right\}_{t=1}^{T}$ be a sequence of $T$ examples for the $k^{th}$ task where $x_k^{(t)} \in \mathbb{R}^d$, $y_k^{(t)} \in \{-1, +1\}$ and $\|x_k^{(t)}\|_2 \leq R$, $\forall t \in [T]$. Let $C$ be a positive constant and let $\alpha$ be some predefined parameter in $[0, 1]$. Let $\{w_k^*\}_{k \in [K]}$ be any arbitrary vectors where $w_k^* \in \mathbb{R}^d$ and its hinge loss on the examples $\left(x_k^{(t)}, y_k^{(t)}\right)$ and $\left(x_j^{(t)}, y_j^{(t)}\right)_{j \neq k}$ are given by $\ell_{kk}^{(t)*} = \left(1 - y_k^{(t)} \langle x_k^{(t)}, w_k^* \rangle\right)_+$ and $\ell_{kj}^{(t)*} = \left(1 - y_j^{(t)} \langle x_j^{(t)}, w_k^* \rangle\right)_+$, respectively.

*If $\{S_k\}_{k \in [K]}$ is presented to OSMTL algorithm, then $\forall k \in [K]$ we have*

$$\sum_{t \in [T]} \left(\ell_{kk}^{(t)} - \ell_{kk}^{(t)*}\right) \leq \frac{1}{2C\alpha} \|w_k^*\|^2 + \frac{(1-\alpha)T}{\alpha}\left(\ell_{kk}^{(t)*} + \max_{j \in [K], j \neq k} \ell_{kj}^{(t)*}\right) + \frac{CR^2 T}{2\alpha} \tag{12}$$

Notice when $\alpha \to 1$, the above reduces to the perceptron mistake bound [17].

**Corollary 2.** *Let* $\alpha = \frac{\sqrt{T}}{1+\sqrt{T}}$ *and* $C = \frac{1+\sqrt{T}}{T}$ *in Theorem 1, we have*

$$\sum_{t\in[T]} (\ell_{kk}^{(t)} - \ell_{kk}^{(t)*}) \leq \sqrt{T}\left(\frac{1}{2}\|w_k^*\|^2 + \ell_{kk}^{(t)*} + \max_{j\in[K],j\neq k} \ell_{kj}^{(t)*} + 2R^2\right) \tag{13}$$

Proofs are given in the supplementary. Theorem 1 and Corollary 2 have several implications:

1. Quality of the bound depends on both $\ell_{kk}^{(t)*}$ and the maximum of $\{\ell_{kj}^{(t)*}\}_{j\in[K],j\neq k}$. In other words, the worst-case regret will be lower if the $k^{th}$ true hypothesis $w_k^*$ can well distinguish training examples in both the $k^{th}$ task itself as well as those in all the other tasks.

2. Corollary 2 indicates the difference between the cumulative loss achieved by our algorithm and by any fixed hypothesis for task $k$ is bounded by a term growing sub-linearly in $T$.

3. Corollary 2 provides a principled way to set hyperparameters to achieve the sub-linear regret bound. Specifically, recall $\alpha$ quantifies the self-concentration of each task. Therefore, $\alpha = \frac{\sqrt{T}}{1+\sqrt{T}} \overset{T\to\infty}{\longrightarrow} 1$ implies for large horizon it would be less necessary to rely on other tasks as available supervision for the task itself is already plenty; $C = \frac{1+\sqrt{T}}{T} \overset{T\to\infty}{\longrightarrow} 0$ suggests diminishing learning rate over the horizon length.

## 3 Experiments

We evaluate the performance of our algorithm under batch and online settings. All reported results in this section are averaged over 30 random runs or permutations of the training data. Unless otherwise specified, all model parameters are chosen via 5-fold cross validation.

### 3.1 Benchmark Datasets

We use three datasets for our experiments. Details are given below:

**Landmine Detection**[3] consists of 19 tasks collected from different landmine fields. Each task is a binary classification problem: landmines $(+)$ or clutter $(-)$ and each example consists of 9 features extracted from radar images with four moment-based features, three correlation-based features, one energy ratio feature and a spatial variance feature. Landmine data is collected from two different terrains: tasks 1-10 are from highly foliated regions and tasks 11-19 are from desert regions, therefore tasks naturally form two clusters. Any hypothesis learned from a task should be able to utilize the information available from other tasks belonging to the same cluster.

**Spam Detection**[4] We use the dataset obtained from ECML PAKDD 2006 Discovery challenge for the spam detection task. We used the task B challenge dataset which consists of labeled training data from the inboxes of 15 users. We consider each user as a single task and the goal is to build a personalized spam filter for each user. Each task is a binary classification problem: spam $(+)$ or non-spam $(-)$ and each example consists of approximately $150K$ features representing term frequency of the word occurrences. Since some spam is universal to all users (e.g. financial scams), some messages might be useful to certain affinity groups, but spam to most others. Such adaptive behavior of user's interests and dis-interests can be modeled efficiently by utilizing the data from other users to learn per-user model parameters.

**Sentiment Analysis**[5] We evaluated our algorithm on product reviews from amazon. The dataset contains product reviews from 24 domains. We consider each domain as a binary classification task. Reviews with rating $> 3$ were labeled positive $(+)$, those with rating $< 3$ were labeled negative $(-)$, reviews with rating $= 3$ are discarded as the sentiments were ambiguous and hard to predict. Similar to the previous dataset, each example consists of approximately $350K$ features representing term frequency of the word occurrences.

We choose 3040 examples (160 training examples per task) for *landmine*, 1500 emails for *spam* (100 emails per user inbox) and 2400 reviews for *sentiment* (100 reviews per domain) for our experiments.

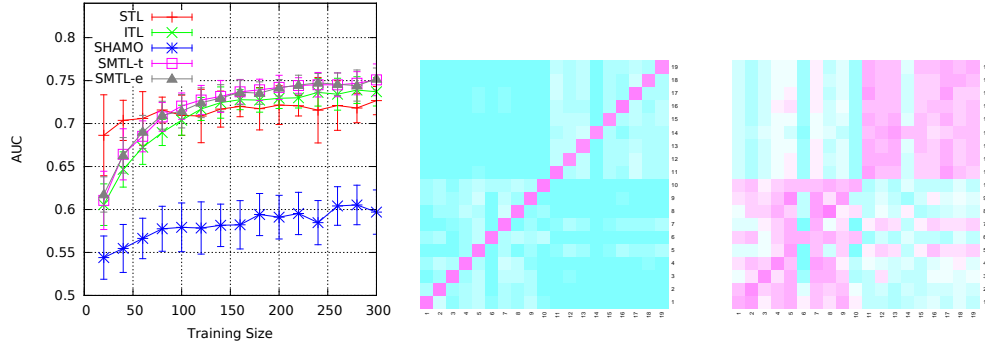

Figure 1: Average $AUC$ calculated for compared models (left). A visualization of the task relationship matrix in *Landmine* learned by *SMTL-t* (middle) and *SMTL-e* (right). The probabilistic formulation of *SMTL-e* allows it to discover more interesting patterns than *SMTL-t*.

Note that we intentionally kept the size of the training data small to drive the need for learning from other tasks, which diminishes as the training sets per task become large. Since all these datasets have a class-imbalance issue (with few $(+)$ examples as compared to $(-)$ examples), we use average Area Under the ROC Curve ($AUC$) as the performance measure.

## 3.2 Batch Setting

Since the main focus of this paper is online learning, we briefly conduct an experiment on landmine detection dataset for our batch learning to demonstrate the advantages of learning from shared data. We implement two versions of our proposed algorithm with different updates: *SMTL-t* (SMTL with thresholding updates) where $p_{kj}^{(t+1)} \propto (\lambda - \ell_{kj}^{(t)})_+$[6] and *SMTL-e* (SMTL with exponential updates) as in Algorithm 1. We compare our *SMTL\** with two standard baseline methods for our batch setting: Independent Task Learning (ITL)—learning a single model for each task and Single Task Learning (STL)—learning a single classification model for pooled data from all the tasks. In addition we compare our models with *SHAMO*, which is closest in spirit with our proposed models. We select the value for $\lambda$ and $\alpha$ for *SMTL\** and $M$ for *SHAMO* using cross validation.

Figure 1 (left) shows the average $AUC$ calculated for different training size on *landmine*. We can see that the baseline results are similar to the ones reported by Xue et. al [3]. Our proposed algorithm (*SMTL\**) outperforms the other baselines but when we have very few training examples (say 20 per task), the performance of STL improves as it has more examples than the others. Since $\eta$ depends on the loss incurred on the data from related tasks, this loss-based measure can be unreliable for a small training sample size. To our surprise, *SHAMO* performs worse than the other models which tells us that assuming two tasks are exactly same (in the sense of hypothesis) may be inappropriate in real-world applications. Figure 1 (middle & left) show the task relationship matrix $\eta$ for *SMTL-t* and *SMTL-e* on *landmine* when the number of training instances is 160 per task.

## 3.3 Online Setting

To evaluate the performance of our algorithm in the online setting, we use all three datasets (*landmine*, *spam* and *sentiment*) and compare our proposed methods to 5 baselines. We implemented two variations of Passive-Aggressive algorithm (*PA*) [18]. *PA-ITL* learns independent model for each task and *PA-ONE* builds a single model for all the tasks. We also implemented the algorithm proposed by Dekel et. al for online multi-task learning with shared loss (*OSGL*) [6]. These three baselines do not exploit the task-relationship or the data from other tasks during model update. Next, we implemented two online multi-task learning related to our approach: *FOML* – initializes $\eta$ with fixed weights [8], Online Multi-Task Relationship Learning (*OMTRL*) [9] – learns a task covariance matrix along with task parameters. We could not find a better way to implement the online version of the SHAMO algorithm, since the number of shared hypotheses or clusters varies over time.

Table 1: Average performance on three datasets: means and standard errors over 30 random shuffles.

| Models | Landmine Detection | | | Spam Detection | | | Sentiment Analysis | | |
|---|---|---|---|---|---|---|---|---|---|
| | AUC | nSV | Time (s) | AUC | nSV | Time (s) | AUC | nSV | Time (s) |
| PA-ONE | 0.5473 (0.12) | 2902.9 (4.21) | 0.01 | 0.8739 (0.01) | 1455.0 (4.64) | 0.16 | 0.7193 (0.03) | 2350.7 (6.36) | 0.19 |
| PA-ITL | 0.5986 (0.04) | 618.1 (27.31) | 0.01 | 0.8350 (0.01) | 1499.9 (0.37) | 0.16 | 0.7364 (0.02) | 2399.9 (0.25) | 0.16 |
| OSGL | 0.6482 (0.03) | 740.8 (42.03) | 0.01 | 0.9551 (0.007) | 1402.6 (13.57) | 0.17 | 0.8375 (0.02) | 2369.3 (14.63) | 0.17 |
| FOML | 0.6322 (0.04) | 426.5 (36.91) | 0.11 | 0.9347 (0.009) | 819.8 (18.57) | 1.5 | 0.8472 (0.02) | 1356.0 (78.49) | 1.20 |
| OMTRL | 0.6409 (0.05) | 432.2 (123.81) | 6.9 | 0.9343 (0.008) | 840.4 (22.67) | 53.6 | 0.7831 (0.02) | 1346.2 (85.99) | 128 |
| OSMTL-t | **0.6776** (0.03) | 333.6 (40.66) | 0.18 | 0.9509 (0.007) | 809.5 (19.35) | 1.4 | 0.9354 0.01 | 1312.8 (79.15) | 2.15 |
| OSMTL-e | 0.6404 (0.04) | 458 (36.79) | 0.19 | **0.9596** (0.006) | 804.2 (19.05) | 1.3 | **0.9465** (0.01) | 1322.2 (80.27) | 2.16 |

Table 1 summarizes the performance of all the above algorithms on the three datasets. In addition to the $AUC$ scores, we report the average total number of support vectors (*nSV*) and the CPU time taken for learning from one instance (*Time*). From the table, it is evident that *OSMTL\** outperforms all the baselines in terms of both $AUC$ and *nSV*. This is expected for the two default baselines (*PA-ITL* and *PA-ONE*). We believe that *PA-ONE* shows better result than *PA-ITL* in *spam* because the former learns the global information (common spam emails) that is quite dominant in spam detection problem. The update rule for *FOML* is similar to ours but using fixed weights. The results justify our claim that making the weights adaptive leads to improved performance.

In addition to better results, our algorithm consumes less or comparable CPU time than the baselines which take into account inter-task relationships. Compared to the *OMTRL* algorithm that recomputes the task covariance matrix every iteration using expensive SVD routines, the adaptive weights in our are updated independently for each task. As specified in [9], we learn the task weight vectors for *OMTRL* separately as $K$ independent perceptron for the first half of the training data available (*EPOCH*=0.5). *OMTRL* potentially looses half the data without learning task-relationship matrix as it depends on the quality of the task weight vectors.

It is evident from the table that algorithms which use loss-based update weights $\eta$ (*OSGL*, *OSMTL\**) considerably outperform the ones that do not use it (*FOML*,*OMTRL*). We believe that loss incurred per instance gives us valuable information for the algorithm to learn from that instance, as well as to evaluate the inter-dependencies among tasks. That said, task relationship information does help by learning from the related tasks' data, but we demonstrate that combining both the task relationship and the loss information can give us a better algorithm, as is evident from our experiments.

We would like to note that our proposed algorithm *OSMTL\** does exceptionally better in *sentiment*, which has been used as a standard benchmark application for domain adaptation experiments in the existing literature [19]. We believe the advantageous results on *sentiment* dataset implies that even with relatively few examples, effectively knowledge transfer among the tasks/domains can be achieved by adaptively choosing the (probabilistic) inter-task relationships from the data.

## 4 Conclusion

We proposed a novel online multi-task learning algorithm that jointly learns the per-task hypothesis and the inter-task relationships. The key idea is based on smoothing the loss function of each task w.r.t. a probabilistic distribution over all tasks, and adaptively refining such distribution over time. In addition to closed-form updating rules, we show our method achieves the sub-linear regret bound. Effectiveness of our algorithm is empirically verified over several benchmark datasets.

## Acknowledgments

This work is supported in part by NSF under grants IIS-1216282 and IIS-1546329.

## Footnotes

[2]It is recommended to set $\alpha \propto \frac{\sqrt{T}}{1+\sqrt{T}}$ and $C \propto \frac{1+\sqrt{T}}{T}$, as suggested by Corollary 2.

[3] http://www.ee.duke.edu/~lcarin/LandmineData.zip

[4] http://ecmlpkdd2006.org/challenge.html

[5] http://www.cs.jhu.edu/~mdredze/datasets/sentiment

[6]Our algorithm and theorem can be easily generalized to other types of updating rules by replacing exp in (6) with other functions. In latter cases, however, $\boldsymbol{\eta}$ may no longer have probabilistic interpretations.

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
