[Supplementary Material]

# Supplementary Material for Adaptive Smoothed Online Multi-Task Learning

**Keerthiram Murugesan**[*]
Carnegie Mellon University
kmuruges@cs.cmu.edu

**Hanxiao Liu**[*]
Carnegie Mellon University
hanxiaol@cs.cmu.edu

**Jaime Carbonell**
Carnegie Mellon University
jgc@cs.cmu.edu

**Yiming Yang**
Carnegie Mellon University
yiming@cs.cmu.edu

## Theoretical Proofs

### Proof of Theorem 1

*Proof.* Define $\Delta_k^{(t)} \stackrel{def}{=} \|w_k^{(t)} - w_k^*\|^2 - \|w_k^{(t+1)} - w_k^*\|^2$.

We can first upper bound $\sum_{t \in T} \Delta_k^{(t)}$ via $\sum_{t \in [T]} \Delta_k^{(t)} = \sum_{t \in [T]} \|w_k^{(t)} - w_k^*\|^2 - \|w_k^{(t+1)} - w_k^*\|^2 = \|w_k^{(0)} - w_k^*\|^2 - \|w_k^{(T+1)} - w_k^*\|^2 \leq \|w_k^*\|^2$.

We further notice any non-zero $\Delta_k^{(t)}$ can be lower-bounded via

$$\Delta_k^{(t)} = \|w_k^{(t)} - w_k^*\|^2 - \|w_k^{(t)} - C\sum_{j \in [K]^+} \eta_{kj}^{(t)} \ell_{kj}^{(t)'} - w_k^*\|^2 \tag{1}$$

$$= 2Cw_k^{(t)} \sum_{j \in [K]^+} \eta_{kj}^{(t)} \ell_{kj}^{(t)'} - 2Cw_k^* \sum_{j \in [K]^+} \eta_{kj}^{(t)} \ell_{kj}^{(t)'} - C^2 \big\| \sum_{j \in [K]^+} \eta_{kj}^{(t)} \ell_{kj}^{(t)'} \big\|_2^2 \tag{2}$$

$$\geq 2C \sum_{j \in [K]^+} \eta_{kj}^{(t)} \big(\ell_{kj}^{(t)} - 1\big) - 2C \sum_{j \in [K]^+} \eta_{kj}^{(t)} \big(\ell_{kj}^{(t)*} - 1\big) - C^2 \big\| \sum_{j \in [K]^+} \eta_{kj}^{(t)} y_j^{(t)} x_j^{(t)} \big\|_2^2 \tag{3}$$

$$= 2C \sum_{j \in [K]^+} \eta_{kj}^{(t)} \ell_{kj}^{(t)} - 2C \sum_{j \in [K]^+} \eta_{kj}^{(t)} \ell_{kj}^{(t)*} - C^2 \big\| \sum_{j \in [K]^+} \eta_{kj}^{(t)} y_j^{(t)} x_j^{(t)} \big\|_2^2 \tag{4}$$

$$\geq 2C\eta_{kk}^{(t)} \ell_{kk}^{(t)} - 2C \sum_{j \in [K]^+} \eta_{kj}^{(t)} \ell_{kj}^{(t)*} - C^2 \Big( \sum_{j \in [K]^+} \eta_{kj}^{(t)} \|x_j^{(t)}\|_2 \Big)^2 \tag{5}$$

$$\geq 2C\eta_{kk}^{(t)} \big(\ell_{kk}^{(t)} - \ell_{kk}^{(t)*}\big) - 2C \sum_{j \in [K]^+, j \neq k} \eta_{kj}^{(t)} \ell_{kj}^{(t)*} - C^2 R^2 \tag{6}$$

$$\geq 2C\alpha \big(\ell_{kk}^{(t)} - \ell_{kk}^{(t)*}\big) - 2C(1-\alpha)\ell_{kk}^{(t)*} - 2C \sum_{j \in [K]^+, j \neq k} \eta_{kj}^{(t)} \ell_{kj}^{(t)*} - C^2 R^2 \tag{7}$$

$$= 2C\alpha \big(\ell_{kk}^{(t)} - \ell_{kk}^{(t)*}\big) - 2C(1-\alpha)\Big(\ell_{kk}^{(t)*} + \sum_{j \in [K]^+, j \neq k} \eta_{kj}^{(t)} \ell_{kj}^{(t)*}\Big) - C^2 R^2 \tag{8}$$

$$\geq 2C\alpha \big(\ell_{kk}^{(t)} - \ell_{kk}^{(t)*}\big) - 2C(1-\alpha)\Big(\ell_{kk}^{(t)*} + \max_{j \in [K]^+, j \neq k} \ell_{kj}^{(t)*}\Big) - C^2 R^2 \tag{9}$$

---

[*]Both student authors contributed equally.

Combining the aforementioned upper and lower bound over $\sum_{t \in [T]} \Delta_k^{(t)}$, we have

$$\sum_{t \in [T]} \left( \ell_{kk}^{(t)} - \ell_{kk}^{(t)*} \right) \leq \frac{1}{2C\alpha} \| w_k^* \|^2 + \frac{(1-\alpha)T}{\alpha} \left( \ell_{kk}^{(t)*} + \max_{j \in [K]^+, j \neq k} \ell_{kj}^{(t)*} \right) + \frac{CR^2 T}{2\alpha} \qquad (10)$$

$$\square$$

**Proof of Corollary 2**

*Proof.* By setting $\alpha = \frac{\sqrt{T}}{1+\sqrt{T}}$ and $C = \frac{1+\sqrt{T}}{T}$, we have

$$\sum_{t \in [T]} \left( \ell_{kk}^{(t)} - \ell_{kk}^{(t)*} \right) \leq \frac{\sqrt{T}}{2} \| w_k^* \|^2 + \sqrt{T} \left( \ell_{kk}^{(t)*} + \max_{j \in [K]^+, j \neq k} \ell_{kj}^{(t)*} \right) + \frac{(1+\sqrt{T})^2}{2\sqrt{T}} R^2 \qquad (11)$$

$$\leq \frac{\sqrt{T}}{2} \| w_k^* \|^2 + \sqrt{T} \left( \ell_{kk}^{(t)*} + \max_{j \in [K]^+, j \neq k} \ell_{kj}^{(t)*} \right) + 2\sqrt{T} R^2 \qquad (12)$$

$$= \sqrt{T} \left( \frac{1}{2} \| w_k^* \|^2 + \ell_{kk}^{(t)*} + \max_{j \in [K]^+, j \neq k} \ell_{kj}^{(t)*} + 2R^2 \right) \qquad (13)$$

Asymptotically, the average regret of our algorithm w.r.t the best predictor $w^*$ in hindsight goes to $0$. Since our algorithm depends on $C$ and $\alpha$, our algorithm needs to know the value of $T$. We can get rid of the dependence of our regret bound on $T$ using the *doubling trick*. $\square$

## Relationship to Domain Adaptation and Life-long Learning

Multi-task learning has been studied in part under a related research topic, *Domain Adaptation* (DA) [1] under different assumptions. There are several key differences between those methods and ours: i) While DA tries to find a *single* hypothesis that works well for both the source and the target data, this paper finds a hypothesis for each task by adaptively leveraging related tasks. ii) It is a typical assumption in DA that the source domains are label-rich and the target domains are label-scarce. However, we are more interested in the scenario where there is a large number of tasks with very few examples available for each task. iii) DA uses predefined uniform weights or weights induced from VC-convergence theory during training, while our method allows cross-task weights to dynamically evolve in an adaptive manner.

The proposed online method is significantly different from lifelong learning (*ELLA* [2]). Unlike our online learning setting where the data from each task arrives in an online fashion, in lifelong learning, task arrives sequentially. At any time-step, the online learner either receives a subset of data for previously solved task or a completely new task.