[Reviews · NeurIPS 2016]

Reviewer 1

Summary

The paper presents a multi-tasks learning approach which encourages individual tasks to learn a model which would predict well on related tasks. The learning also learns the relation among tasks in the form of a row-stochastic matrix. Both batch and an online algorithms are provided based on closed form solutions for efficient computation.

Qualitative Assessment

I liked the paper. It presents an interesting approach and shows promising results. The paper is written clearly and easy to follow. Following are a few issues in a descending order of importance: 1. One of the major strength of the paper is the presentation of the closed form solutions (eq. (6), (8) & (10)) which enable efficient algorithms. I find the paper relevant to a wide community and I am not sure all readers would master the proximal operator or mirror descent. I think the paper would benefit a lot if the solutions were presented or at least pointed to a relevant reference where the full solution exists. 2. The regret is sub-linear only if the alpha and C are set to specific values dependent on T. I find this dependence on T confusing. The authors themselves state that in regular online setting where data is continually streamed T=inf, in which case the individual tasks have sufficient samples to learn on their own. This makes the regret analysis and the sub-linear derivation somewhat redundant, doesn't it? To me it seems the interesting case is when alpha is set to a value << 1 and sharing is heavily encouraged. In the experiment section line 272 it is implied that these values are set dynamically based on the growing 't', this is interesting, decreasing the incentive to share as more samples are provided. 3. Following my previous remark. It would be interesting to see performance as a function of alpha? 4. In the batch approach the Max entropy is used. This seems like a very strong prior on the nature of relations among tasks, encouraging a solution with uniform attention. This seems like a significant prior which is not discussed/analyzed (only mentioned in line 143). It would be interesting to add a discussion showing the effect of this prior on the learnt matrix and possibly consider alternative regularization terms. 5. I found the following reference: hierarchical regularization cascade for joint learning Zweig et al. very relevant to the paper. It also deals with multi-task online and batch learning and provides a similar regret bound. They are able to share information among tasks while avoiding explicitly finding or modeling the hierarchy. Clearly there is a lot of benefit of finding the hierarchy and it would be very interesting to compare both methods and see if the explicit discovery of the hierarchy comes at a cost of classification performance. 6. domain adaptation: I don't think the discussion about domain adaptation is critical to the paper, but if it remains I suggest to change it. They define DA based on a single citetation. Not all domain adaptation follow into this narrow definition, e.g see papers in https://people.eecs.berkeley.edu/~jhoffman/domainadapt/ . 7. I find the usage of the term "adaptive" confusing because it indicates that the model is able to adapt to changes in tasks distributions and relation among tasks. But this is not the case as evident in the regret analysis which considers a single optimal classifier over the entire sequence. I don't think the presented approach is tailored for non-stationary distributions anymore then regular online learning. 8. Typos: line 67: "has also has" line 4-5: "whether learners focus on their specific tasks or on jointly address all tasks."

Confidence in this Review

3-Expert (read the paper in detail, know the area, quite certain of my opinion)


Reviewer 2

Summary

This paper presents an online multi-task learning algorithm with probabilistic inter-task relationships, which can be formulated as a composite of an identity matrix and a row-stochastic matrix. Batch formulation and online formulation are proposed. Regret bound is analyzed.

Qualitative Assessment

This paper presents an online multi-task learning algorithm with probabilistic inter-task relationships, which can be formulated as a composite of an identity matrix and a row-stochastic matrix P. Although the row-stochastic matrix P leads to a probability distribution over all tasks for each specific task, P is not constrained to be symmetric. A concern here is whether the asymmetric P can cause any inconsistency of task relationships. One of the contributions of the paper is the regret bound, however I am not sure whether this bound is useful since it depends on values of w*_k, l^(t)*_kk and so on, which can be quite loose. The experimental results are promising.

Confidence in this Review

2-Confident (read it all; understood it all reasonably well)


Reviewer 3

Summary

Authors proposed a method to do Online Multi-Task Learning. In their work, they: - Choose dataset from one domain (ex: spam detection, and 2 others), and split the data into multiple tasks (ex: each "user" is a task), and learn a per-user spam detector while leveraging the global data. - Formulate a loss objective (equations 1 and 2), that makes each parameter vector "w_k" do well on its own task loss "L_k" but at the same time, do well on other task losses "L_j" even for "j != k". - They model matrix \eta with entry \eta_{kj} being the "similarity" between tasks k and j. In particular, the total loss for selecting w_k is \sum_j \eta_{kj} L_j(w_k) -- in other words, w_k should incur low loss for other tasks where \eta_{kj} is high. They propose to define \eta as: \eta = \alpha * I_K + (1 - \alpha) P; where \alpha is a scalar in [0, 1]; I_K is KxK identity matrix; and P is a row-stochastic matrix. - They discuss alternatives where \alpha and P are fixed (for example SHAMO [1]) but they also explain their algorithm, which trains P and varies (increments) \alpha over time, where the choice of \alpha is justified in section 2.5 (Regret Bound). - They propose an "Online Formulation" where the parameters evolve smoothly over time: They evolve to minimize the loss, but also to minimize the changes of parameters from previous iteration/example. Their "smoothness" is achieved by: * minimizing the L2 norm of delta "w", this is similar [20], though they *dont* cite this similarity (maybe it is obvious??) * minimizing the change in P, this is novel in online learning as far as my little knowledge goes: where they suggest to minimize the KL divergence: KL(P_k^(t+1) || P_k^(t)) - They show that their model performs better than others on 3 multi-task, binary datasets.

Qualitative Assessment

The paper is technically sound, with some minor flaws that are explained below. Overall, I am on the "edge" of deciding if this should be accepted or rejected as a Poster. The largest factor contributing to my "reject" thought is the incompleteness of the experiments, as explained below. Nonetheless, I will provide lots of things that confused me and can be improved: Problem Setup: 1- L19, and in other places: you claim "online learning", yet you assume that all K tasks emit examples at the same time step, which inherently invalidates your motivation: now data from all tasks is equal in quantity. You mention on L153 that the assumption can be relaxed with "added complexity". I *think* you should mention how one can train without that assumption. Language / math correctness: 2- You mention on L23, L156, and elsewhere, that your model only updates when it makes a mistake. This does not agree with the math you have. In particular, your hinge-loss emits updates *even* if your prediction is on the correct side of the hyper-plane but if it was within "1" (margin) away from the hyperplane. 3- You mention probability simplex and use \Delta^{K-1}. What does probability simplex mean? I tried googling it and failed to find a justifying answer. I know about the tetrahedron geometric simplex but I bit it is not it. Did you just mean a "distribution" [i.e. positive entries summing to 1]? And why set the dimensions to K-1? Is it because the diagonal entry in P is set to zero since there is an Identity matrix in equation 3 which fills the diagonal? 4- Line 139, equation (4), the notation got confusing! What is \eta_{kj}(p_k) ? I am assuming that it is the (k, j)-th entry with the given p_k. Regardless if my assumption is correct, please explain notation right after you present it the first time. 5- In equation 6, my immediate response was a shock that there is no \alpha in the update rule. I should calculate the derivative myself perhaps. Unfair comparison: 6- For your argument on multi-task learning, You should experiment with OSMTL-ONE and OSMTL-ITL, to show that your data-sharing loss is effective. You only compare the -ONE and -ITL using the PA algorithm, which is not equivalent to OSMLT, and therefore it is not clear if your wins come from the "smooth" online updates (PA is 'aggressive') or if they come from the data sharing. You don't want to over-sell the multi-task (MT) aspect if the wins comes from the "smooth" (S). If somehow PA-ONE is equivalent to OSMLT-ONE, please make it clear in the paper. Incompleteness: 7- The online experimental setting is not clear. In particular, do you go over the training data just once? What is the size of the test data? was it held-out during training? Please complete this information. If the datasets have Train : Test splits and you have used them, please mention so.

Confidence in this Review

2-Confident (read it all; understood it all reasonably well)


Reviewer 4

Summary

This paper proposes an online multi-task learning algorithm that has a probabilistic interpretation and efficient updating rules. It also includes support for varying individual weights between fitting individual tasks or generalizing to multiple tasks. The authors provide a theoretical analysis showing that the proposed method has sublinear regret. Empirical evaluations on a variety of multi-task learning data sets show that this approach outperforms other methods.

Qualitative Assessment

This paper was clearly written and provided a nice approach with sublinear regret. The presentation is excellent overall, and the experiments compared to many alternative methods with good analysis. I'm slightly concerned of the problem setting with the direct reuse of weight vectors between task models, since that restricts the diversity of tasks that the method can handle, but this same setting is used by many other MTL methods. Although the experiments compared to multiple online MTL methods, the authors should also have compared to ELLA (Ruvolo et al, ICML'13), which is an online version of Daume's GO-MTL algorithm (which outperforms OMTRL by a good bit). According to the ICML'13 paper, ELLA obtains around a 0.77 AUC on Landmine, for example (which outperforms the proposed method). However, the experiment setup is different (the tasks are received consecutively, not the data), so it is hard to judge whether ELLA would beat the proposed method without re-running the experiment. But, ELLA does support incremental learning of each task, so the comparison should be made before publication. line 188: typo "su-blinear"

Confidence in this Review

2-Confident (read it all; understood it all reasonably well)


Reviewer 5

Summary

This paper investigates the online multi-task learning problem and proposes a new scheme targeting at jointly learning both the per-task model parameters and the inter-task relationships. The core of this method lies in the parametrization of task relationship matrix and featuring the probabilistic interpretation. Experiments on three benchmark datasets validate its effectiveness and efficiency.

Qualitative Assessment

This paper proposes a new online multi-task learning method, which can jointly learn both the per-task model parameters and the inter-task relationships in an adaptive manner. My major comments are as follows. 1) This paper focuses on the online version of the typical multi-task problem. In this paper, at some round t all tasks are processed with the incremental data points. I think this setting is sort of strict. However, there is another setting called lifelong learning, which is very similar to the online multi-task learning. I do not see any discussion about the differences between them in the related work. As far as I am concerned, lifelong learning (e.g. ELLA: An Efficient Lifelong Learning Algorithm, ICML2013) can also learn the task parameters and the task-relationship matrix iteratively. More discussions are needed. 2) The novelty of this paper is also not strong. The only novel part in this paper is the parametrization of the task relationship matrix. However, this implementation is kind of plain. Besides, I wonder whether there is overlapping between \eye{I}_k and the diagonal elements of P. 3) The whole paper focuses on the binary classification problem and utilizes hinge loss as the loss function. I think it might be not very general to represent all multi-task problems, such as regression or any other loss function choices. 4) In line 108, it claims that “regularizer r prevents degenerate cases”. I think more explanations should be given since it is important in your paper. 5) The difference from SPL is not well-illustrated. In my point of view, the Unfocused update of the proposed method is quite similar to SPL. 6) What about the update order of each task in Eq.(7)? Besides, why is the negative entropy in batch version replaced by KL divergence in online setting? More explanations are needed. 7) This paper claims that the authors are more interested in the scenario where there is a large volume of tasks with few examples available. Nevertheless, in Experiment section the used datasets are very small. It would be better and more convincing if larger datasets are involved in the experimental validation.

Confidence in this Review

2-Confident (read it all; understood it all reasonably well)


Reviewer 6

Summary

This paper addresses multi-task learning via pairwise task relationships using the shared data. Their model has a probabilistic interpretation of the pairwise weights. They give batch and online algorithm, the latter being the main focus. The online version has bounded and sublinear regret proved in Theorem 1 and corollary 1. Experiments show improvement over single task learning as well as some previous approaches to multi task learning, most significantly in the sentiment analysis task. I recommend this paper for a strong accept.

Qualitative Assessment

The idea of pairwise interactions is very natural and the derived theoretical results are nice. It is interesting that the interactions are not constrained to be symmetric in this work, making the probabilistic interpretation a bit difficult. I believe the paper is accessible including the proofs in supplementary material.

Confidence in this Review

2-Confident (read it all; understood it all reasonably well)